# Biodegradable Poly-ε-Caprolactone Scaffolds with ECFCs and iMSCs for Tissue-Engineered Heart Valves

**DOI:** 10.3390/ijms23010527

**Published:** 2022-01-04

**Authors:** Georg Lutter, Thomas Puehler, Lukas Cyganek, Jette Seiler, Anita Rogler, Tanja Herberth, Philipp Knueppel, Stanislav N. Gorb, Janarthanan Sathananthan, Stephanie Sellers, Oliver J. Müller, Derk Frank, Irma Haben

**Affiliations:** 1Department of Cardiovascular Surgery, University Hospital Schleswig-Holstein (UKSH), 24105 Kiel, Germany; thomas.puehler@uksh.de (T.P.); jette.seiler@uksh.de (J.S.); anita.rogler@uksh.de (A.R.); tanja.herberth@uksh.de (T.H.); philipp.knueppel@uksh.de (P.K.); Irma.haben@uksh.de (I.H.); 2German Centre for Cardiovascular Research (DZHK), Partner Site Hamburg/Kiel/Lübeck, 20251 Hamburg, Germany; oliver.mueller@uksh.de (O.J.M.); derk.frank@uksh.de (D.F.); 3Stem Cell Unit, Clinic for Cardiology and Pneumology, University Medical Center Göttingen, 37075 Göttingen, Germany; lukas.cyganek@gwdg.de; 4German Center for Cardiovascular Research (DZHK), Partner Site Göttingen, 37075 Göttingen, Germany; 5Department of Functional Morphology and Biomechanics, Zoological Institute, Christian-Albrechts-University of Kiel, 24105 Kiel, Germany; sgorb@zoologie.uni-kiel.de; 6Department of Centre for Heart Valve Innovation, St Paul’s Hospital, University of British Columbia, Vancouver, BC V6T 174, Canada; jsathananthan@providencehealth.bc.ca (J.S.); SSellers@providencehealth.bc.ca (S.S.); 7Department of Cardiology and Angiology, University Hospital Schleswig-Holstein (UKSH), 24105 Kiel, Germany

**Keywords:** tissue engineering, heart valve, ECFCs, iMSCs, PCL nanofibers, biodegradable

## Abstract

Clinically used heart valve prostheses, despite their progress, are still associated with limitations. Biodegradable poly-ε-caprolactone (PCL) nanofiber scaffolds, as a matrix, were seeded with human endothelial colony-forming cells (ECFCs) and human induced-pluripotent stem cells-derived MSCs (iMSCs) for the generation of tissue-engineered heart valves. Cell adhesion, proliferation, and distribution, as well as the effects of coating PCL nanofibers, were analyzed by fluorescence microscopy and SEM. Mechanical properties of seeded PCL scaffolds were investigated under uniaxial loading. iPSCs were used to differentiate into iMSCs via mesoderm. The obtained iMSCs exhibited a comparable phenotype and surface marker expression to adult human MSCs and were capable of multilineage differentiation. EFCFs and MSCs showed good adhesion and distribution on PCL fibers, forming a closed cell cover. Coating of the fibers resulted in an increased cell number only at an early time point; from day 7 of colonization, there was no difference between cell numbers on coated and uncoated PCL fibers. The mechanical properties of PCL scaffolds under uniaxial loading were compared with native porcine pulmonary valve leaflets. The Young’s modulus and mean elongation at F_max_ of unseeded PCL scaffolds were comparable to those of native leaflets (*p* = ns.). Colonization of PCL scaffolds with human ECFCs or iMSCs did not alter these properties (*p* = ns.). However, the native heart valves exhibited a maximum tensile stress at a force of 1.2 ± 0.5 N, whereas it was lower in the unseeded PCL scaffolds (0.6 ± 0.0 N, *p* < 0.05). A closed cell layer on PCL tissues did not change the values of F_max_ (ECFCs: 0.6 ± 0.1 N; iMSCs: 0.7 ± 0.1 N). Here, a successful two-phase protocol, based on the timed use of differentiation factors for efficient differentiation of human iPSCs into iMSCs, was developed. Furthermore, we demonstrated the successful colonization of a biodegradable PCL nanofiber matrix with human ECFCs and iMSCs suitable for the generation of tissue-engineered heart valves. A closed cell cover was already evident after 14 days for ECFCs and 21 days for MSCs. The PCL tissue did not show major mechanical differences compared to native heart valves, which was not altered by short-term surface colonization with human cells in the absence of an extracellular matrix.

## 1. Introduction

Tissue engineering offers a unique opportunity by providing a living valve that is capable of growth and biological integration [1]. Despite the enormous progress made in the development of tissue-engineered heart valves, a clinically relevant and commonly used product has not yet been realized. Leaflets of native human heart valves with their orthotropic nature consist of proteoglycans, highly organized collagen network, elastin fibers (ECM), and valve interstitial cells (VIC). They are surrounded by an outer layer of specialized endothelial cells (valvular endothelial cells = VEC). The adult endothelial progenitor cells used in cardiovascular research originate from the bone marrow, circulate in the peripheral blood, and contribute to neovascularization [2]. These so-called endothelial colony forming cells (ECFCs) are easily isolated from the blood [3] and are suitable for therapeutic use [4,5].

MSCs are very similar to VICs that have antithrombogenic and immunosuppressive properties [6,7,8]. Since 2007, human MSCs have been generated artificially from induced pluripotent stem cells (iPSCs) [9]. iPSCs have been produced by artificial reprogramming of non-pluripotent somatic cells [9,10] and a high medical potential as autologous, i.e., patient-specific cells [11]. iPSCs can be differentiated into MSCs in a further step shown by numerous groups in recent years (summarized in [12]). This artificial generation of MSCs allows efficient and numerous multiplication of multipotent cells with a greater expansion capacity [13] and diminished tumorigenic potential [14,15]. 

In addition, especially for heart valves, the composition and structural organization of the extracellular matrix is the most important factor for mechanical function [16]. 

Poly-ε-caprolactone (PCL) is one of the favored synthetic biodegradable biomaterials [17] in tissue engineering due to its high processability and advantageous mechanical properties [18]. Its fabrication into scaffolds of nanofibers that mimic the three-dimensional (3D) structure of the ECM and promote cell adhesion and proliferation is particularly promising [19]. The use of PCL scaffolds seeded with MSCs has been investigated in the area of bone replacement for several years. 

However, large and irregularly extended cell aggregates have been observed, which can reduce both the scaling potential and the differentiation potential of cells and the secretion of paracrine factors [20]. Nevertheless, since the PCL is biodegradable, it is favored that the cells, used for colonization, form an ECM to give the heart valve its desired mechanical properties.

The novelty of this study is that we investigated the attachment, growth, and colonization of randomly oriented PCL scaffolds mimicking native decellularized ECM tissue with ECFCs and MSCs for the generation of tissue-engineered heart valves. Human ECFCs were isolated from the blood and human iMSCs were successfully differentiated from induced pluripotent stem cells (iPSCs) via mesoderm in a new fast two-phase protocol. 

In addition, it has been evaluated whether the coating of the PCL matrix appears to have a significant effect on mid-term viability and the final outcome of colonization. 

Furthermore, the mechanical properties of the PCL scaffolds were correlated to native heart valve leaflets in order to analyze whether short-term colonization of the cells had already an effect on the mechanical PCL properties.

## 2. Results

### 2.1. Characterization of Human ECFCs

Isolation of human peripheral blood derived ECFCs was successfully established. The characterization of human ECFCs isolated from peripheral blood occurred through the formation of a cobblestone-like morphology (Figure 1a), the capacity to form capillary-like structures (Figure 1b), and the analysis of the surface marker profile (Figure 1c). In culture, appearance of ECFC colonies took 21–28 days. The ECFCs exhibited the following a surface marker profile: CD31^+^, CD34^−^, CD45^−^, CD146^+^, KDR^+^, and CD309^+^.

### 2.2. Generation and Characterization of Human iMSCs

iMSCs were successfully generated from two human iPS cell lines derived from skin fibroblasts using a feeder-free protocol (Figure 2a). To confirm the generation of iMSCs, the transition of iPSCs into the MSC-like phenotype (Figure 2b) and gene expression (Figure 2c) was monitored. iPSCs showed expression of pluripotency markers Sox2 and Oct4, which were lost after mesoderm induction of the cells. As brachyury is considered to be one of the best markers of early mesoderm [21], its expression was analyzed on day 5 and day 6. Subsequently, MSCs were induced by incubation with MSC growth medium. Resulting cells showed no gene expression for all investigated markers (day 8). 

To verify their MSC-like-phenotype, trilineage differentiation capacity was demonstrated by culture in adipogenic, chondrogenic, and osteogenic induction media (Figure 2d). Mineral deposition by iMSCs cultured in osteogenic induction medium indicated early stages of bone formation. Fat globules were detected in iMSC culture grown in adipogenic induction medium indicating differentiating into adipocytes. Chondrocyte pellet of iMSCs cultured in osteogenic induction medium indicated formation of cartilage. In addition, the surface marker profile of iMSCs was characterized by the expressions of CD31, CD34, CD44, CD45, CD90, CD146, and CD166 using flow cytometry (Figure 2e). iMSCs exhibited positive expression of CD44 (96.33%), CD90 (99.05%), CD146 (91.57%), and CD166 (97.94%). The endothelial marker CD31, the hematopoietic stem cell marker CD34 and the hematopoietic marker CD45 were only detected with an expression of 1.01%, 1.5%, and 4.12%, respectively, suggesting that iMSCs have a similar surface marker profile to human MSCs.

### 2.3. Morphological Characterization of PCL Nanofibers

SEM images were examined to analyze the difference between the 20 µm thick fiber layer on the PCL plates and the loose PCL tissues with a layer thickness of 100 µm. In both cases, the randomly crossing nanofibers appeared, but only the PCL tissues showed a 3D structure that resembled the ECM (Figure 3). 

### 2.4. Coating of PCL Plates for Seeding with ECFCs and MSCs

Cell adhesion, proliferation, and distribution on the PCL fibers were analyzed by fluorescence microscopy and SEM. ECFCs adhered and proliferated on PCL fibers and formed a closed cell layer on day 28 (Figure 4). Small gaps in the cell cover were associated with artefacts of sample processing. Even though the highest cell number was observed on day 7 (Figure 5), after two weeks of culturing, the cells were evenly distributed and more elongated than on the days before. 

To analyze the effect of coated PCL fibers on cell adhesion and proliferation, ECFCs and MSC were seeded onto uncoated Matrigel-, gelatin-, or collagen I-coated 24-well PCL plates, and evaluated over a period of 14 and 21 days, respectively (Figure 5). While coating the fibers with Matrigel significantly increased the number of living ECFCs on day 4, the coating had no significant effect on cell numbers at the later time points on days 7 and 14 (Figure 5a). 

The same was true for MSCs, as they adhered and proliferated on uncoated PCL fibers and formed a nearly closed cell layer on day 21 (Figure 6). Starting from day 14, they appeared evenly distributed and elongated. Again, the cell numbers were significantly increased on gelatin-coated fibers at days 1 and 4, but not after 7 and more days of culturing. MSCs proliferated more slowly than ECFCs. Nevertheless, they expanded until day 14 and cell numbers remained stable until day 21 (Figure 5b). Since no significant differences in cell numbers were found on the differently coated PCL plates after day 14 for both ECFCs and MSCs, uncoated PCL fibers were used for the following experiments.

### 2.5. Cell Seeding of PCL Tissues

To analyze the influence of the 3D scaffolds on the proliferation and distribution of the cells, ECFCs were cultured on PCL tissues for a period of 14 days. In SEM, the morphology during the growth of ECFCs on uncoated PCL tissue showed how the distribution of cells became more uniform as the colonization progressed (Figure 7). In addition, a change in the size and shape of the cells was observed as an increasingly closed cell layer was formed, comparable to the results on plate. In contrast to the results on plate, a closed cell layer with a smooth surface was already visible on day 14. Again, the cells were uniformly arranged within the cell layer and they did not exhibit any orientation in any particular direction. Fluorescence analysis indicated a surface distribution of the cells; migration into the PCL tissue could not be observed after 14 days.

To further analyze the effects of fiber sizes, iMSCs were seeded on uncoated PCL tissue with varying fiber sizes (300 and 700 nm) and analyzed two weeks later. Colonization of PCL fibers with iMSCs demonstrated good adhesion, growth, and alignment of cells on the scaffold. The comparison between 300 and 700 nm fiber PCL tissues showed a very dense cell layer after 15 days in both cases (Figure 8). However, iMSC seeded on 300 nm PCL fibers showed fewer gaps, close cell-cell contacts and, thus, a nearly confluent cell coverage after 15 days.

Thus, compared to human iMSCs, human ECFCs were shown to grow faster on the PCL fibers, forming a closed cell layer after only 14 days on PCL tissue. Accordingly, the growth data of the cells on PCL plates also showed slower growth of the MSCs compared to the ECFCs (Figure 5). Nevertheless, a confluent cell layer could be achieved with the porcine MSCs after 21 days.

### 2.6. Mechanical Characterization of Native Porcine Heart Valves and PCL Scaffolds

To determine the mechanical properties of the PCL scaffolds, the behavior of the biodegradable PCL scaffolds under uniaxial loading was investigated and compared to native porcine pulmonary valves. The Young’s modulus, maximum tensile stress force (F_max_), and mean elongation at F_max_ were considered in the analysis (Table 1).

The tissue of the porcine pulmonary valves exhibited a Young’s modulus of 4.4 ± 3.4 MPa and a mean elongation at F_max_ of 108.8 ± 49.6%. Similarly, PCL scaffolds achieved a Young’s modulus of 3.9 ± 0.8 MPa and a mean elongation at F_max_ of 143.5 ± 34.8%. Colonization of PCL scaffolds with human ECFCs or iMSCs did not significantly alter these properties compared to native heart valves (ECFCs: 3.6 ± 1 MPa and 129.4 ± 10.2% and iMSCs: 2.7 ± 0.5 MPa and 141.1 ± 5.7%, respectively). In terms of these mechanical properties, the PCL scaffolds were equivalent to native heart valves and did not deteriorate with cell colonization. Of note, the values of native heart valves showed a wide standard deviation, which is most likely due to the biological nature of the material.

In contrast, the native heart valves exhibited a maximum tensile stress force (F_max_) of 1.2 ± 0.5 N, whereas it was lower in the PCL scaffolds (0.6 ± 0.1 N), indicating that PCL is a mildly weaker material in only this aspect, which might fail more quickly. After 14 days of seeding with cells, the PCL tissue showed similar values of F_max_ as unseeded PCL tissues (ECFCs: 0.6 ± 0.1 N; iMSCs: 0.7 ± 0.1 N). Accordingly, the values also differed from those of native heart valves.

## 3. Discussion

A major shortcoming for current valve prostheses is the absence of cells required for active repair and remodeling of the scaffold. The use of blood-derived ECFCs and iMSCs is particularly appealing because they can both be obtained in a noninvasive manner, greatly expanded in vitro, allowing colonization of PCL scaffolds with autologous cells to enable biological integration of a living heart valve. 

In this study, the growth of ECFCs and iMSCs on PCL nanofiber scaffolds were studied. ECFCs are particularly well suited for tissue-engineered heart valve colonization because they closely resemble mature vascular endothelial cells [22]. Their high doubling time, as also reported in the literature [23,24], allows rapid expansion on the PCL fibers. Consistent with the literature, no toxic effect of PCL on cells cultured thereon was observed at any time point [18,25].

PCL, as a hydrophobic polymer, does not exhibit optimal properties for interaction with cells, which is a disadvantage compared to other biomaterials. It is known that cells adhere more easily to hydrophilic surfaces [26] and better adhesion is associated with faster cell spreading [27]. Numerous works have therefore combined PCL with other polymers with more favorable biological properties, improving cell adhesion, and viability on scaffolds [18,28,29]. Accordingly, coating of the fibers with matrix proteins (i.e., Matrigel or collagen I) resulted in significantly increased cell number at an early time point. 

However, the coating does not appear to have a significant effect on mid-term viability and the final outcome of colonization, indicating that the coating is mainly important for adhesion and initial proliferation of the cells. From day 7 of colonization on, there was no difference between cell numbers on coated and uncoated PCL fibers detectable, suggesting uncoated PCL fibers are also suitable as a matrix for colonization with ECFCs. 

A successful two-phase protocol based on the timed use of differentiation factors for efficient differentiation of iPSCs into iMSCs has been developed. Firstly, the induction of mesodermal cells was initiated by the addition of specific growth factors. MSCs were then formed in a second step. The derived iMSCs met the minimal criteria defined for the use of MSCs in cell therapy [30]. Thus, this rapid method of differentiation of iPSCs into iMSCs may be advantageous for the use of MSC in regenerative medicine. Colonization of PCL fibers with MSCs also resulted in good adhesion, proliferation, and distribution with tight compaction of cells with cell–cell contacts. Compared to ECFC, MSC showed a lower growth rate, but again a completely closed cell layer was observed after 21 days. When comparing the fiber thickness of the PCL tissue, the lower fiber thickness of 300 nm in diameter seemed to have a positive impact on the cell growth of the iMSCs as previously reported [31]. Similar to ECFCs, the fiber coatings showed only significantly improved initial adhesion of MSCs. From day 7 onwards, however, coating no longer had any effect on the number of living cells. This emphasizes that due to the different cell types with their distinct functions and locations, different scaffolds should be considered for tissue engineering.

Heart valves provide unidirectional blood flow through the heart by opening and closing in a circular fashion, which requires exceptional mechanical properties. Poly-ε-caprolactone is one of the favored synthetic biomaterials as it combines many desirable properties such as biocompatibility, biodegradability, mechanical strength and flexibility [17]. Its biocompatibility and strength with good results in cell infiltration [19,32] makes it particularly interesting for the production of implantable long-term prostheses. PCL has already been approved by the Food and Drug Administration (FDA) for specific uses in the human body [33]. 

Several groups are working on the colonization of such PCL scaffolds produced by electrospinning for cardiovascular use. It was shown that different cell types require different chemical modifications of the fibers as well as topographical properties of the scaffolds for adhesion [34,35,36]. PCL and PCL-based fibrous scaffolds have been used in heart valve tissue engineering, and fiber diameters of those scaffolds are in nanoscale and/or microscale [25,37,38]. Due to the 3D arrangement of the randomly oriented fibers of PCL, not only adhesion, but also migration of the cells into the tissue is possible. 

In addition, PCL tissues can be integrated into steal or nitinol stents to produce functional heart valved stents. The PCL scaffold investigated exhibited similar values of Young’s modulus and mean elongation at F_max_, indicating that the material has comparable elasticity and strain to failure. However, a smaller force was sufficient to rupture the PCL, resulting in a weaker material. To obtain information on whether the PCL tissue exhibit altered mechanical properties after colonization with ECFCs and iMSCs, PCL tissues covered with cells were analyzed. Cell colonization in a biochemically stimulated environment had no negative influence on the mechanical properties investigated in this study compared to native heart valves. 

Unfortunately, comparison of mechanical properties with the literature is limited due to differences in sample sizes, evaluated sizes, and settings. The fluorescence images of the colonization experiments indicated no migration of MSCs into the PCL tissue after 14 days. The unchanged mechanical properties of the PCL tissue also support this conclusion. 

Nevertheless, the formation of an ECM (elastin and collagen) is fundamental for the durability and longevity of heart valves [39]. A variety of in vitro studies analyzed different biocompatible scaffolds seeded with autologous cells to generate a collagen-rich ECM [38]. Confluent MSCs were shown to be able to form and deposit collagen when ascorbic acid was added, but short-term culture time resulted in insufficient formation of MSC-derived ECM [40,41]. 

Here, MSCs showed a completely closed cell layer only after 21 days, whereas tensile tests were performed after a colonization of 14 days. Moreover, the experiments were performed under static conditions, which does not correspond to the natural environment for the cells of a heart valve. Biophysical stimuli, such as those found in heart valves, affect MSCs, e.g., migration and differentiation [42]. Consistent with this, mononuclear cells were shown to migrate into decellularized aortic valves after only 3 days under dynamic conditions. However, MSCs again proved to be the dominant cell population only after 3 weeks [43]. Nevertheless, in vivo, implantation of decellularized aortic valves reseeded with MSCs isolated from bone marrow showed promising results in the sheep model [44].

Limitations: further analysis with longer colonization times, dynamic colonization, and achieved migration of cells into the material is required. An optimal colonization strategy needs to be developed for the successive seeding of MSC’s and ECFC’s. MSC-generated ECM should be tested in tensile tests for its effect on mechanical properties. Moreover, the fabrication of a PCL scaffold in the form of a tricuspid heart valve for seeding in the bioreactor under physical conditions is necessary prior further in vivo experiments.

However, our hypothesis is that colonization with a combination of iMSCs and ECFCs is a promising concept. Studies have shown a positive interaction between MSCs and human umbilical endothelial cells [45]. The combined intravenous administration of MSCs and ECFCs potentiated the vasculogenic response in vivo and enhanced the functional repair of infarct-damaged tissue [46]. Furthermore, it was shown that the release of proangiogenic growth factors by MSCs stimulated the proliferation and formation of capillary-like structures of ECFCs [47]. In turn, ECFCs secrete different signaling molecules that stimulate differentiation of the MSCs [48]. Cultivation of umbilical cord endothelial cells with iPSC–MSCs already promoted vascularization in vitro and subsequent bone regeneration in vivo [49]. 

Thus, the colonization of biodegradable PCL with autologous iMSCs and ECFCs in combination may provide a living, remodeling, and completely natural heart valve with comparable mechanical properties.

## 4. Material & Methods

### 4.1. Ethics Statement

The collection of peripheral blood for ECFC isolation was approved by the local ethical committee (D464/16). All patients provided written informed consent before the recruitment after receiving a full explanation of the study. The generation and biobanking of the iPSC lines is covered by the ethics vote (21 January, 2011 and 10 September 2015) from the ethics commission of the University Medical Center of Göttingen, Germany. The study procedures were approved by the ethics committee of the Medical Faculty of Medicine of the Kiel University, Germany (approval number: D522/16).

### 4.2. ECFC Isolation and Culture

Human ECFCs were isolated from peripheral blood according to a published protocol [50] with small modification: blood was diluted 1:1 with Dulbecco’s phosphate buffered saline (PBS; Thermo Fisher Scientific, Waltham, MA, USA) supplemented with 2% fetal bovine serum (FBS; Thermo Fisher Scientific, USA) and layered on Lymphoprep (STEMCELL Technologies, Vancouver, BC, Canada). Buffy coat mononuclear cells were plated on T75 cell culture flasks with collagen type I coating (Corning, Corning, NY, USA) cultured in endothelial cell basal medium 2 (EBM-2 medium; Lonza, Switzerland) supplemented with EGM-2-SingleQuots (Lonza), 10% FBS, and 1% penicillin–streptomycin (PS; Thermo Fisher Scientific). EBM-2 medium was changed three times per week. After 48 h, non-adherent cells were removed by rinsing with PBS. After 7 days, cells were transferred to 24-well plates coated with collagen type I. ECFC colonies with characteristic cobblestone morphology appeared after 10 to 16 days and were expanded on 6-well plates with collagen type I coating.

### 4.3. Characterization of ECFCs

Matrigel tube formation assay (as previously described [51]) and flow cytometry were performed to characterize the cells in passage 5. Tube formation was examined every two hours using phase contrast microscopy (Zeiss, Jena, Germany). 

For flow cytometry, cells were washed with PBS and resuspended in staining solution consisting of 2% FBS in PBS. For flow cytometry, cells were washed with PBS and resuspended in staining solution consisting of 2% FBS in PBS. Cell samples were separately labeled on ice with optimal dilution of conjugated monoclonal antibodies (all from BioLegend, San Diego, CA, USA) against Brilliant Violet 421 mouse anti-human CD31 (clone WM59), fluorescein isothiocyanate (FITC) mouse anti-human CD34 (clone 561), FITC mouse anti-human CD45 (clone HI30), FITC mouse anti-human CD146 (clone P1H12), and phycoerythrin (PE) mouse anti-human CD309 (clone 7D4-6). For analysis, cells were dissolved in 200 μL PBS containing 1% paraformaldehyde (Sigma-Aldrich, St. Louis, MI, USA). Nonspecific fluorescence was determined by incubation of cell aliquots with isotype-matched antibodies. Flow cytometry was performed on a BD LSRFortessa (Becton Dickinson, Franklin Lakes, NJ, USA) using the software BD FACSDiva (Becton Dickinson) and Flowing Software 2 (Perttu Terho, University of Turku, Turku, Finland) for further evaluation.

### 4.4. iPSC Culture

The human iPS cell lines ipWT1.1 and ipWT1.3 were generated from primary human fibroblasts derived from skin biopsy of a clinically silent healthy donor [52] and kindly provided by Dr. Lukas Cyganek of the Stem Cell Unit, University Medical Center Göttingen, Germany. In brief, the hiPSC line was generated using the integration-free episomal 4-in-1 CoMiP reprogramming plasmid (no. 63726; Addgene, Watertown, MA, USA) with the reprogramming factors OCT4, KLF4, SOX2, and c-MYC and short hairpin RNA against p53, as described previously with modifications [53]. iPSCs were cultured on Corning Matrigel membrane matrix (Thermo Fisher Scientific, Waltham, MA, USA) in the feeder-free culture medium mTeSR Plus (STEMCELL Technologies, BC, Canada) in a humidified atmosphere at 37 °C and 5% CO_2_. 

### 4.5. Generation of hiPSC-MSCs via Mesoderm

iPSCs were seeded without feeder cells on Matrigel at 5000 cells/cm^2^ and cultured in mTeSR Plus supplemented with ROCK pathway inhibitor Y-27632 (Enzo Life Sciences, Lörrach, Germany). After 24 h, the medium was changed to STEMdiff Mesoderm Induction Medium (STEMCELL Technologies) and replenished daily for four days. On day 5 of differentiation, medium was replaced with StemXVivo MSC Expansion Medium (R&D Systems, Minneapolis, MN, USA). After 48 h, cells were dissociated with 0.25% trypsin/EDTA (Thermo Fisher Scientific) and seeded onto gelatin-coated plates at 1  ×  10^4^ cells/cm^2^ in MSC expansion medium. Medium was replenished every other day, and cells were propagated to 80% confluence in a humidified atmosphere at 37 °C and 5% CO_2_. The differentiated cells derived from these culture conditions were termed iPSC-derived MSCs (iMSCs) and expanded. Cells of an early passage were used for characterization. For routine expansion, cells were plated at 5  ×  10^4^ cells/cm^2^ onto Matrigel or uncoated culture dishes (starting with P3) and maintained in MSC growth medium. MSC growth medium consisted of Dulbecco’s modified Eagle’s medium-high glucose (DMEM-HG; Thermo Fisher Scientific), 10% FBS, and 1% PS.

### 4.6. Gene Expression Analysis

Total RNA from iPSCs, mesoderm cells and iMSCs was harvested using TRIzol reagent (Life Technologies, Carlsbad, CA, USA) and reverse transcribed to cDNA (2 ng of total RNA) with GoScript reverse transcriptase (Promega, Madison, WI, USA). cDNA samples were subjected to PCR to analyze gene expression of the pluripotency markers OCT3/4 and SOX2, the mesoderm marker Brachyury, and B2M, used as an internal reference. Oligonucleotide primers synthesized by Eurofins Scientific; Luxembourg are listed in Table 2.

### 4.7. Multilineage Differentiation Assays

Induction of osteogenic, adipogenic, of chondrogenic differentiation was performed as previously descripted [54]. Briefly, for osteogenesis and adipogenesis, iMSCs were cultured in the respective medium, and every three to four days, cells were washed with PBS and the respective medium was replaced. After 21 days, cultures were fixed in formalin and stained for 20 min in alizarin red solution to visualize calcium deposits or with Oil red O (both Sigma-Aldrich, St. Louis, MO, USA) in isopropanol for detection of lipid accumulation, respectively. Cells were examined under inverted microscope for evidence of bone or fat differentiation. 

For chondrogenesis, iMSCs were cultured in the respective medium. Twenty-four hours after the onset of micromass formation, the culture medium was replaced every three to four days. After 21 days, the chondrocyte pellet was fixed in formalin, embedded in paraffin, and 5-μm sections were stained with 1% toluidine blue/1% sodium borate. 

### 4.8. Flow Cytometric Analysis of iMSCs 

Detection of cell surface antigen profile of iMSCs was performed identical to flow cytometry of ECFCs. Monoclonal antibodies (all BioLegend) used: allophycocyanin (APC) mouse anti-human CD34, FITC mouse anti-human CD44, APC/Cy7 mouse anti-human CD45, PE/Cy7 mouse anti-human CD90, FITC mouse anti-human CD146, and PE mouse anti-human CD166. 

### 4.9. Porcine MSC Culture

Porcine MSCs were included in the colonization of PCL plates as in vivo evaluation of tissue-engineered heart valves is being performed in the porcine model and porcine iPSCs, generated via a non-integrating method, are currently not available. For in vivo evaluation in the porcine model, MSCs derived from bone marrow can be used instead as an autologous cell source. The porcine bone marrow mesenchymal stem cells (Cell Biologics, Chicago, IL, USA) were thawed and expanded in MSC growth medium on uncoated culture dishes in a humidified atmosphere at 37 °C and 5% CO_2_. 

### 4.10. Electrospun PCL Scaffolds 

Randomly oriented biodegradable electrospun PCL scaffolds, purchased from Nanofiber Solutions, USA, with a fiber size of 300 and 700 nm in diameter were used for seeding experiments. A random orientation of the fibers mimics the 3D nanofibrous extracellular matrix found throughout the body and increases the surface area and pores, which facilitates adhesion of higher cell numbers. Moreover, 24-well plates with attached PCL-nanofibers (20-µm fiber layer) and loose PCL scaffolds (PCL tissue; 700 nm fiber diameter) with a layer thickness of 100 µm were used.

### 4.11. Cell Seeding of PCL Culture Plates

To investigate whether prior coating of PCL fibers will optimize colonization, 5.3 × 10^3^ cells/cm^2^ ECFCs and 2.6 × 10^4^ cells/cm^2^ pMSCs were seeded on coated PCL 24-well plates. For ECFCs, plates were coated with either Matrigel (1:120 in PBS) or collagen I (1:71.5 in acetic acid); for pMSCs, plates were coated with either Matrigel or gelatin (0.1% gelatin (Merck, Darmstadt, Germany). Uncoated plates were used for each cell type as reference. Seeding of PCL plates was performed as described before [55]. On days 1, 4, 7, and 14 for ECFCs and on days 1, 4, 7, 14, 21 for pMSCs, plates were analyzed using a NyOne image cytometer (Synentec, Elmshorn, Germany) and scanning electron microscopy.

### 4.12. Fluorescent Staining and Automated Cell Imaging

Seeded 24-well PCL plates were analyzed using a NyOne image cytometer. Cells were stained with Hoechst 33342, CellTracker Red CMTPX (both Thermo Fisher Scientific), and CellTox Green (Promega, USA). Plates were blocked with medium supplemented with 10% goat serum (PAN Biotech, Aidenbach, Germany), for 20 min at 37 °C. Hoechst and CellTracker Red were dissolved according to manufacturer’s instructions and the staining solution was incubated for 30 min at 37 °C. Plates were washed with PBS, and CellTox Green was added according to manufacturer’s instructions. After 15 min of staining at room temperature, the plates were washed with PBS and were analyzed using a NyOne image cytometer. 

For evaluation of the automated cell count, cells stained with Hoechst and CellTracker Red, but not CellTox Green, were defined as living cells. Cells stained with CellTox Green were defined as dead cells. For each day and coating, an unseeded well was used as control, and values for living and dead cells were subtracted from those of seeded wells of the same coating. 

### 4.13. Cell Seeding of PCL Tissue

PCL tissues were cut into 5 × 15 mm pieces and incubated in 6-well plates, with medium at 37 °C for 30 min. After removal of the medium, ECFCs and MSCs were diluted in 10 μL medium and added directly to the wet PCL tissue using a single-channel pipette. Before adding further medium, scaffolds were incubated for 30 min at 37 °C. One side was seeded with 5.2 × 10^3^ cells/cm^2^ on day 0. The next day, the scaffolds were inverted and the other side was seeded with the same number of cells. Samples were colonized in a humidified atmosphere at 37 °C and 5% CO_2_.

### 4.14. SEM Analysis

Unseeded and seeded PCL scaffolds were air dried with 1,1,1,3,3,3-hexamethyldisilazan (HMDS, Roth). To enhance conductivity, a 10-nm thin gold/palladium layer was sputtered on the samples. Scanning electron microscopic micrographs were taken with a Hitachi TM-3000. The surface images were captured at an accelerating voltage of 15.0 kV at different magnifications. 

### 4.15. Dissection of Porcine Pulmonary Heart Valves

Hearts of 15 adult crossbred German swine were obtained from a slaughterhouse at Kiel using aseptic conditions. The hearts were rinsed with lactated Ringer solution and placed in sterile DMEM. The dissection of porcine pulmonary heart valves was performed as described before [56].

### 4.16. Biomechanical Examination

To investigate the biomechanical characteristics of the native leaflets and PCL tissues (seeded for 14 days and unseeded, but soaked in nutrient medium), 5 × 15 mm strips were characterized by uniaxial tensile testing. Tensile tests were performed using a universal testing machine (ZwickRoell Z0.5, Ulm, Germany) with a 5 kN load cell and a constant strain rate of 0.8%/min. Samples were stretched very slowly in radial direction at 2.0 mm/min until complete rupture. Young’s modulus, F_max_ and elongation at break were recorded [57,58]. 

### 4.17. Statistical Analysis

All statistical tests were performed using Prism software (GraphPad Software, San Diego, CA, USA). Results are presented as mean ± SD; *p* values ≤ 0.05 (Student’s *t* test) were considered statistically significant.

## 5. Conclusions

A successful two-phase protocol based on the timed use of differentiation factors for efficient differentiation of human iPSCs into iMSCs was developed: cells were generated using a novel feeder-free differentiation protocol in a fast manner.

The aim of this research project was to analyze PCL tissue as a matrix for tissue-engineered heart valves seeded with human ECFCs and iMSCs. We therefore established a patient-specific cell source for adequate seeding of tissue-engineered constructs. In terms of mechanical properties, the PCL scaffolds were nearly equivalent to native heart valves and did not deteriorate with cell colonization. 

We demonstrated that both human ECFCs and MSCs adhered to uncoated surfaces generated by electrospinning from PCL fibers, and they are therefore suitable for colonization to generate tissue engineered heart valve constructs. 

## Figures and Tables

**Figure 1 ijms-23-00527-f001:**
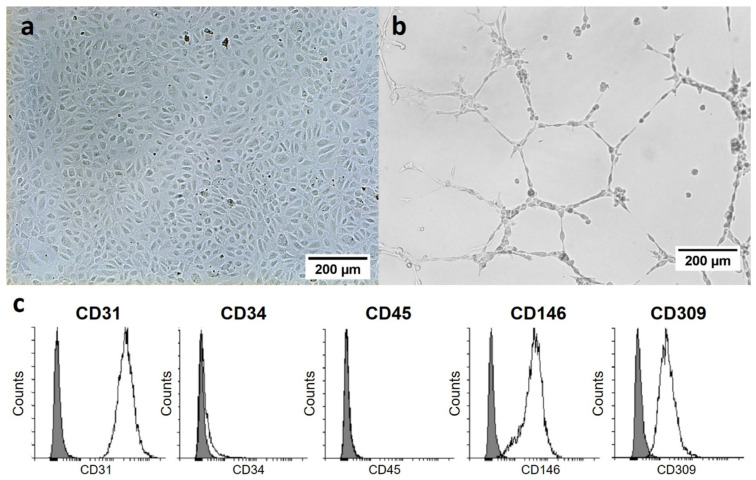
Human ECFCs isolated from peripheral blood. (**a**) Cobblestone-like single-cell layer of the ECFCs on a collagen-coated culture plate. (**b**) Growth of the ECFCs on a base membrane-like matrix leads to the formation of capillary-like structures. (**c**) Flow cytometric analysis of surface markers of ECFCs (white: ECFCs; grey: negative control).

**Figure 2 ijms-23-00527-f002:**
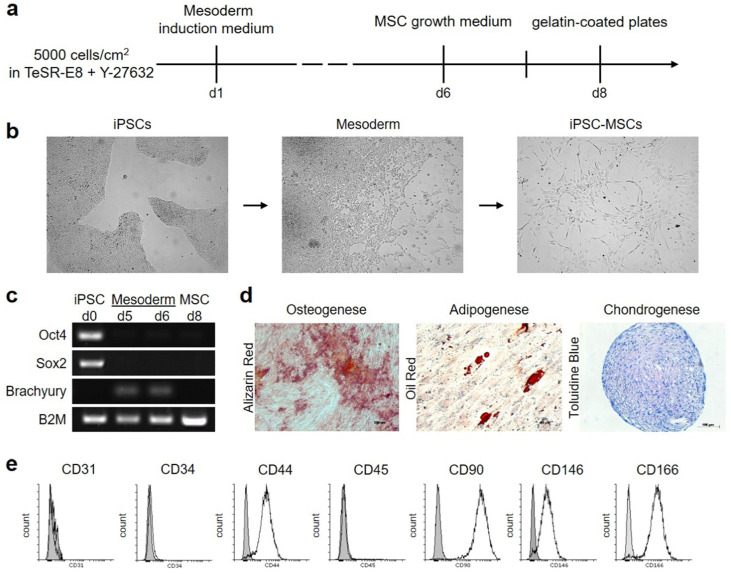
Induction of iPSCs into MSC-like cells. (**a**) Schematic differentiation protocol of iPSCs to iMSCs. (**b**) Light microscopy images demonstrating the cell morphology changes occurring during the development of iPSC to hiPSC-MSCs. Representative cell morphology of iPS cells prior to induction. A representative image of the cell morphology of mesoderm cells after induction with Mesoderm Induction Medium. (**c**) Gene expression of different cell types (Oct4: octamer-binding transcription factor 4, Sox2: sex determining region Y-box 2, B2M: Beta-2-Microglobulin). (**d**) Multilineage differentiation of iMSCs. (**e**) Flow cytometric analysis of surface markers of hiPSC-MSCs (white: iMSCs, grey: negative control).

**Figure 3 ijms-23-00527-f003:**
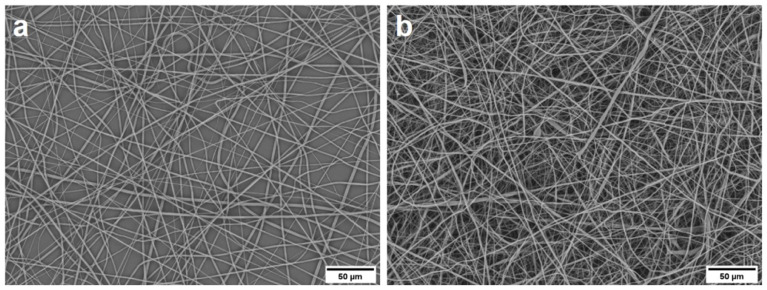
SEM images of the uncoated PCL fiber scaffolds with 700 nm fiber size in diameter of (**a**) the 24-well plate and (**b**) the loose PCL tissue.

**Figure 4 ijms-23-00527-f004:**
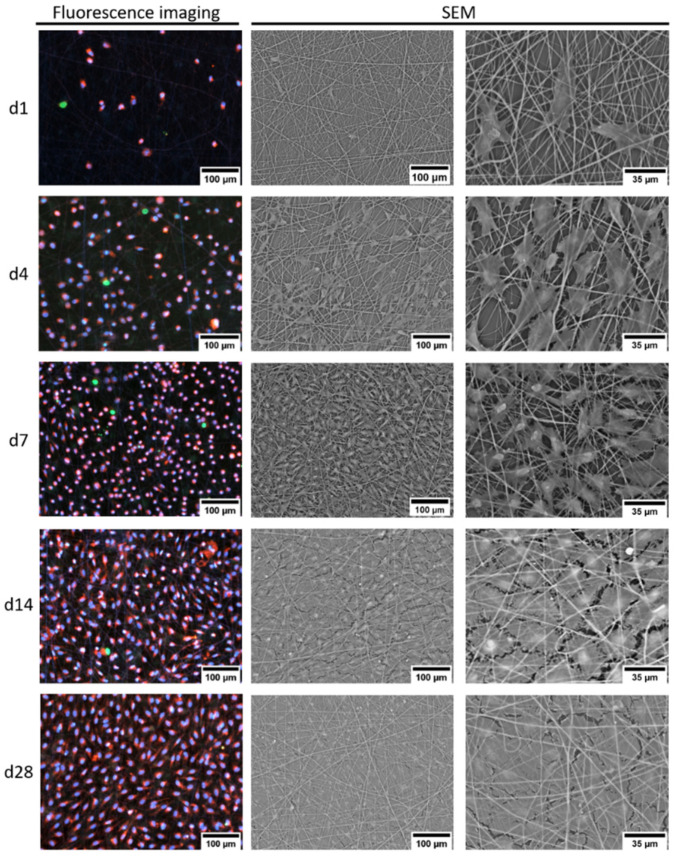
Representative fluorescence and scanning electron microscopy images of the morphology of the growth of ECFCs on uncoated PCL fibers for up to 28 days.

**Figure 5 ijms-23-00527-f005:**
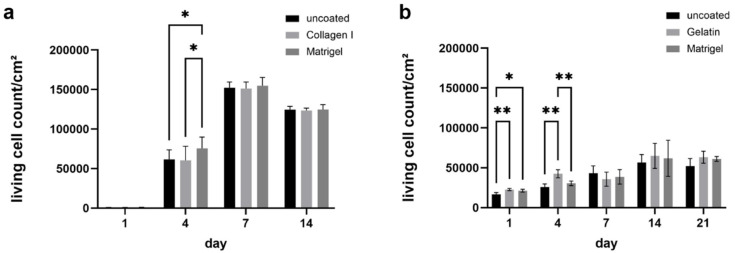
Seeding of 24-well of un- and coated PCL plates with ECFCs or pMSCs. Count of living cells/cm^2^ of (**a**) ECFCs and (**b**) pMSCs. Results are expressed as mean + SD: * *p* < 0.05; ** *p* < 0.001 (Student’s *t* test).

**Figure 6 ijms-23-00527-f006:**
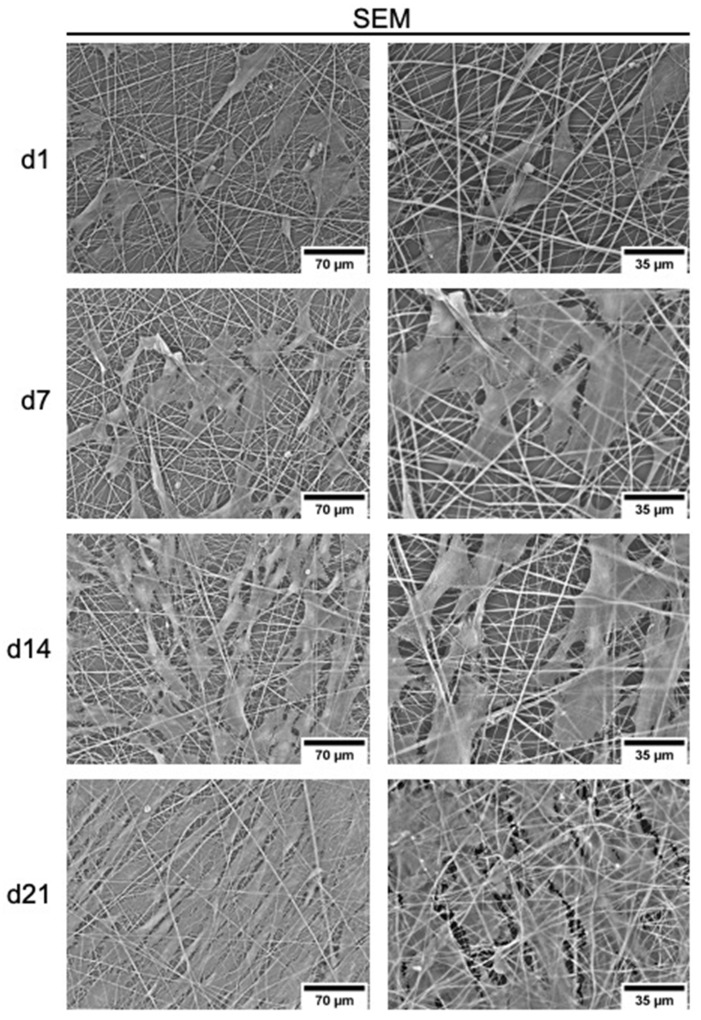
Representative scanning electron microscopy images of the morphology of the growth of MSCs on uncoated PCL fibers for 21 days.

**Figure 7 ijms-23-00527-f007:**
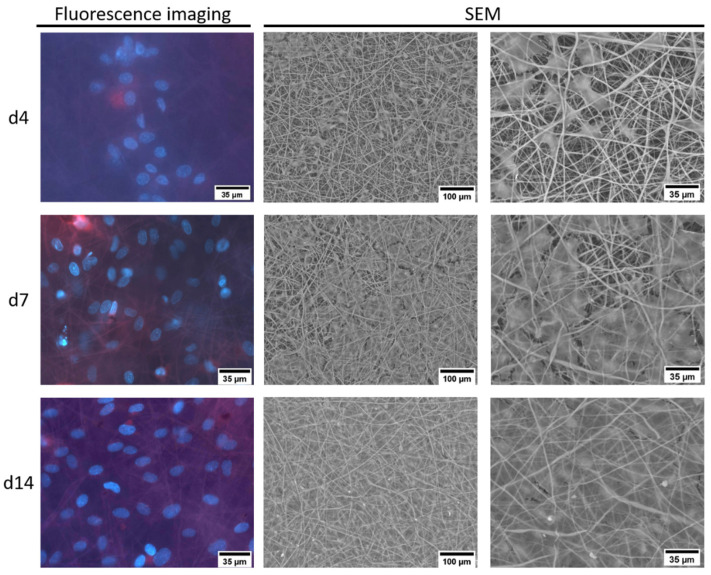
Representative SEM and fluorescence microscopy images of uncoated PCL tissues (700 nm fiber size) seeded with ECFCs for 4, 7, and 14 days.

**Figure 8 ijms-23-00527-f008:**
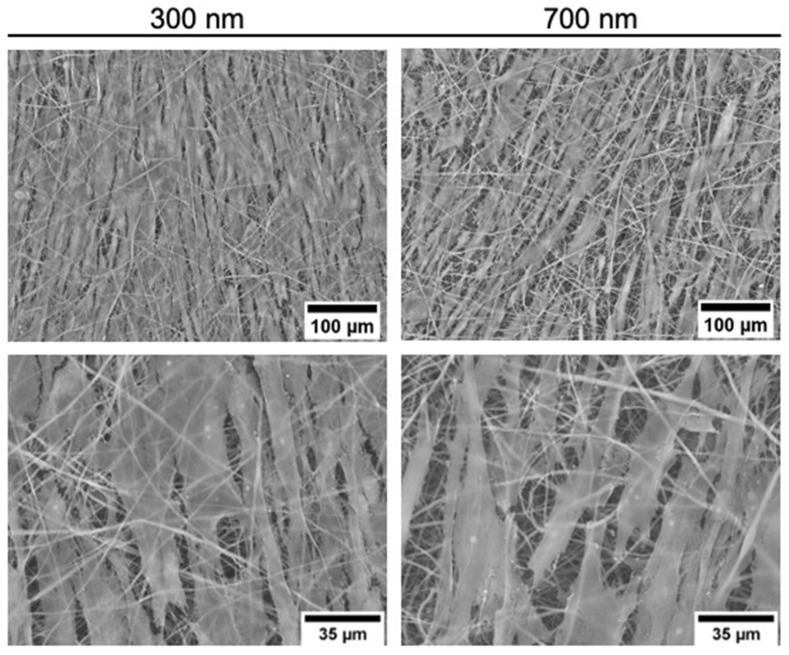
Representative SEM images of uncoated PCL scaffolds (300 and 700 nm) seeded with human iMSCs for 15 days.

**Table 1 ijms-23-00527-t001:** Uniaxial tensile mechanical properties: Young’s modulus, F_max_, and elongation at F_max_ of native porcine pulmonary heart valves, unseeded PCL scaffolds, and PCL scaffolds seeded with iMSCs or hECFCs for 14 days.

	Young’s Modulus (MPa)	F_max_ (N)	Elongation at F_max_ (%)
Native (*n* = 15)	4.4 ± 3.4	1.2 ± 0.5 *	108.8 ± 49.6
PCL (*n* = 10)	3.9 ± 0.8	0.6 ± 0.1 *	143.5 ± 34.8
PCL + ECFCs (*n* = 11)	3.6 ± 1.0	0.6 ± 0.1 *	129.4 ± 10.2
PCL + iMSCs (*n* = 6)	2.7 ± 0.5	0.7 ± 0.1 *	147.1 ± 5.7

Mean values ± SD are shown. Asterisks indicate significant differences of mean values between native heart valves and PCL tissues (unseeded and seeded, respectively): * *p* < 0.05 (Student’s *t* test).

**Table 2 ijms-23-00527-t002:** Sequences of the sense and antisense primers used for RT-PCR.

Genes	Sense Primer	Antisense Primer
B2M	AAG CAG CAT CAT GGA GGT TTG	GAG CTA CCT GTG GAG CAA CC
OCT3/4	AGT AGT CCC TTC GCA AGC C	CCC CCA CAG AAC TCA TAC GG
Brachyury	CAA CCT CAC TGA CGG TGA AAA A	ACA AAT TCT GGT GTG CCA AAG TT
SOX2	AAC CAG CGC ATG GAC AGT T	GCA AAG CTC CTA CCG TAC CA

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
