# Peer review of "Biodegradable Poly-ε-Caprolactone Scaffolds with ECFCs and iMSCs for Tissue-Engineered Heart Valves"

_ijms, 2022, doi:10.3390/ijms23010527_

Round 1

Reviewer 1 Report

Dear authors,

Here are my comments for this paper.

  1. Please add the novelty of the paper with respect to existent literature.
  2. The Introduction section should be extended by adding more examples and references to sustain the presented ideas.
  3. Did you coat the samples before SEM analysis?
  4. What standard was used for mechanical analysis?
  5. Please provinde stress-strain curves.
  6. The number of references is very very low for a scientific paper. You must add more relvant papers, esepciall for the introduction section. The discussion section could benefit from these new references.
  7. Section 2.9 - did you prepare the materials? It seems that the scaffolds were bought from Nanofiber solutions.

Author Response

Q1: Please add the novelty of the paper with respect to the existent literature.

A1: Thank you for your comment: we fully reworked the paper now to point out the novelty of our study and discuss this in more detail with respect to the existent literature. See mainly the Introduction and Discussion sections.  

Q2 : The Introduction section should be extended by adding more examples and references to sustain the presented ideas.

A2: We extended the introduction section and included necessary literature to strengthen our presented ideas of this study.

Q3: Did you coat the samples before SEM analysis?

A3: We analyzed coated and uncoated samples: see Fig. 3-8 with the description of un- or coated samples. Fig. 5 demonstrates the comparison of un- and coated samples.We also discussed this en detail in the end of paragraph 3 and fully in paragraph 4 of the Discussion section.  

Q4: What standard was used for mechanical analysis?

A4: The international standard of the Society of Materials Information was used. Two references describing these techniques in detail are indicated in the Material and Methods section, now. See also references 28 and 29.

Q5: Please provide stress-strain curves.

A5: Uniaxial tensile mechanical properties are provided in detail. Stress strain curves were not provided due to the large amount of figures we already used. They easily can be provided as well.

Q 6: The number of references is very very low for a scientific paper. You must add more relvant papers, espcially for the introduction section. The discussion section could benefit from these new references.

A6: We provided now more references for the Introduction, M&M, and the Discussion section (altogether n=59).

Q7: Section 2.9 - did you prepare the materials? It seems that the scaffolds were bought from Nanofiber solutions.

A7: This is correct. “Randomly oriented biodegradable electrospun PCL scaffolds, purchased from Nanofiber Solutions, USA, with a fiber size of 300 nm and 700 nm in diameter were used for seeding experiments“ as indicated in the Material and Methods section (2.10. Electrospun PCL scaffolds).

Reviewer 2 Report

The work is really interesting and it will be valuable for the literature.

However, a few points are needed to address before it can be accepted for publication

  • Poly-ε-caprolactone should be poly-ε-caprolactone
  • Improve the scale font size for SEM images
  • Fig 3 graph fonts are too small, needed to be improved

Author Response

The work is really interesting and it will be valuable for the literature. However, a few points are needed to address before it can be accepted for publication

Q1: Poly-ε-caprolactone should be poly-ε-caprolactone

A1: We changed the first letter according to your suggestion.

Q2: Improve the scale font size for SEM images

A2: We improved the scale font size for the SEM images.

Q3: Fig 3. graph fonts are too small, needed to be improved

A3: This has been done as suggested.

Reviewer 3 Report

The orthotropic nature of a native pulmonary valve leaflet (which is characterised by a highly organised collagen fibres network) has not sufficiently discussed.  Static colonization of PCL fibers with iPSC-MSCs has been only considered.

The role of mechanically stimuli on PCL scaffolds seeded with iMSCs is missed in the introduction and final discussion. This  aspect is particularly relevant for heart valves leaflets.

Seeded biodegradable scaffolds when cultured in a mechanically (and biochemical) stimulated environment, which induces extracellular matrix production and an additional potential driving force for differentiation, could be finally characterised by orthotropic biomechanical properties. Although the paper is basically correct in the experimental and data analysis, the aspects related to the highly relevant biomechanics differentiation have not been sufficiently emphasized by the Authors in the introduction and in the conclusions. More references should be added regarding this topic.

The Authors should play more attention in using the terms "stress" and "strength".

Author Response

Q1: The orthotropic nature of a native pulmonary valve leaflet (which is characterised by a highly organised collagen fibres network) has not sufficiently discussed.  Static colonization of PCL fibers with iPSC-MSCs has been only considered.

A1: This is now added in the Introduction section in paragraph 1: “Despite the enormous progress in the development of tissue-engineered heart valves a clinically relevant and commonly used product has not yet been realized. Leaflets of native human heart valves with their orthotropic nature consist of proteoglycans, highly organized collagen network, elastin fibers (ECM), and valve interstitial cells (VIC). They are surrounded by an outer layer of specialized endothelial cells (valvular endothelial cells = VEC). The adult endothelial progenitor cells used in cardiovascular research originate from the bone marrow, circulate in the peripheral blood and contribute to neovascularization [2]. These so-called endothelial colony forming cells (ECFCs) are easily isolated from the blood [3] and are suitable for therapeutic use [4,5].”

Q2: The role of mechanically stimuli on PCL scaffolds seeded with iMSCs is missed in the introduction and final discussion. This aspect is particularly relevant for heart valves leaflets.

A2: The following has been added in the Introduction section to underline the mechanically stimuli of PCL:

“In addition, especially for heart valves, the composition and structural organization of the extracellular matrix is the most important factor for mechanical function [16].”

Furthermore, it has been added now the following mechanical stimuli on PCL scaffolds seeded with MSCs in the third paragraph of the Introduction section:

“Poly-ε-caprolactone (PCL) is one of the favored synthetic biodegradable biomaterials [17] in tissue engineering due to its high processability and advantageous mechanical properties [18]. Its fabrication into scaffolds of nanofibers that mimic the three-dimensional (3D) structure of the ECM and promote cell adhesion and proliferation is particularly promising [19]. The use of PCL scaffolds seeded with MSCs has been investigated in the area of ​​bone replacement for several years. However, large and irregularly extended cell aggregates have been observed, which can reduce both the scaling potential and the differentiation potential of cells and the secretion of paracrine factors [20]. Nevertheless, since the PCL is biodegradable, it is favored that the cells, used for colonization, form an ECM to give the heart valve its desired mechanical properties.”

Q3: Seeded biodegradable scaffolds when cultured in a mechanically (and biochemical) stimulated environment, which induces extracellular matrix production and an additional potential driving force for differentiation, could be finally characterised by orthotropic biomechanical properties.

A3: We fully agree: we added the following paragraph in the Discussion section on this topic (see paragraph 11 in the Discussion section): “In addition, PCL tissues can be integrated into stents to produce functional heart valved stents. The PCL scaffold investigated exhibited similar values of Young’s modulus and mean elongation at Fmax, indicating that the material has comparable elasticity and strain to failure. However, a smaller force was sufficient to rupture the PCL, resulting in a weaker material. To obtain information on whether the PCL tissue exhibit altered mechanical properties after colonization with ECFCs and iMSCs, PCL tissues covered with cells were analyzed. Cell colonization in a biochemically stimulated environment had no negative influence on the mechanical properties investigated in this study compared to native heart valves.”

Q4: Although the paper is basically correct in the experimental and data analysis, the aspects related to the highly relevant biomechanics differentiation have not been sufficiently emphasized by the Authors in the introduction and in the conclusions.

A4: Cerainly, the biomechanical differentiation during cell seeding should be mentioned and discussed. This is now added in the Discussion section (12th to 14th paragraph of Discussion section): “Unfortunately, comparison of mechanical properties with the literature is limited due to differences in sample size, evaluated sizes, and settings. The fluorescence images of the colonization experiments indicated no migration of MSCs into the PCL tissue after 14 days. The unchanged mechanical properties of the PCL tissue also support this conclusion. Nevertheless, the formation of an ECM (elastin and collagen) is fundamental for the durability and longevity of heart valves [49]. A variety of in vitro studies analysed different biocompatible scaffolds seeded with autologous cells to generate a collagen-rich ECM [48]. Confluent MSCs were shown to be able to form and deposit collagen when ascorbic acid was added, but short-term culture time resulted in insufficient formation of MSC-derived ECM [51].

Here, MSCs showed a completely closed cell layer only after 21 days, whereas tensile tests were performed after a colonization of 14 days. Moreover, the experiments were performed under static conditions, which does not correspond to the natural environment for the cells of a heart valve. Biophysical stimuli, such as those found in heart valves, affect MSCs, e.g., migration and differentiation [52]. Consistent with this, mononuclear cells were shown to migrate into decellularized aortic valves after only 3 days under dynamic conditions. But MSCs again proved to be the dominant cell population only after 3 weeks [53]. Nevertheless, in vivo, implantation of decellularized aortic valves reseeded with MSCs isolated from bone marrow showed promising results in the sheep model [54].”

Q5: More references should be added regarding this topic.

A5: The list of references is expanded, now (n=59).

Q6: The Authors should play more attention in using the terms "stress" and "strength".

A6: This is true. We added the following paragraph in the Discussion section on this topic (see paragraph 9 of the Discussion section): „Heart valves provide unidirectional blood flow through the heart by opening and closing in a circular fashion which requires exceptional mechanical properties. Poly-ε-caprolactone is one of the favored synthetic biomaterials as it combines many desirable properties such as biocompatibility, biodegradability, mechanical strength and flexibility [17]. Its‘ biocompatibility and strength with good results in cell infiltration [41,42] makes it particularly interesting for the production of implantable long-term prostheses. PCL has already been approved by the Food and Drug Administration (FDA) for specific uses in the human body [43].“

Round 2

Reviewer 1 Report

Dear authors,

It is cleared that your paper has strongly been improved. The Introduction section can still benefit from some improvements, especially the novelty which is shown only in the discussion by comparison to other paper. Also, in the conclusion section, the novelty could be added.

Author Response

Dear Reviewer

As suggested by you: the novelty of this study has been now added in the introduction section (last paragraph), in the conclusion section of the abstract, and in the conclusion of the discussion section (see green sentences).

Only minor flaws were found according to the English language and were corrected. 

If you wish to have a native speaker proofreading again, please let us know (duration 10-14 days).

Kindest, 

Pühler and Lutter. 

Reviewer 3 Report

The paper has been revised according to my suggestions

Author Response

Dear reviewer,

thank you very much for your very valuable former suggestions. These have made the manuscript very important for the readers now.

We have uploaded the manuscript again with responses to reviewer one.

Kindest regards.

Thomas Pühler and Georg Lutter